# GAP: Graph-based Agent Planning with Parallel Tool Use and Reinforcement Learning

**Jiaqi Wu[1], Qinlao Zhao[2], Zefeng Chen[3], Kai Qin[1],**
**Yifei Zhao[1], Xueqian Wang[1], Yuhang Yao[4]**
[1]Tsinghua University    [2]Huazhong University of Science and Technology
[3]National University of Singapore    [4]Carnegie Mellon University
wu-jq24@mails.tsinghua.edu.cn
yuhangya@alumni.cmu.edu

## Abstract

Autonomous agents powered by large language models (LLMs) have shown impressive capabilities in tool manipulation for complex task-solving. However, existing paradigms such as ReAct rely on sequential reasoning and execution, failing to exploit the inherent parallelism among independent sub-tasks. This sequential bottleneck leads to inefficient tool utilization and suboptimal performance in multi-step reasoning scenarios. We introduce **G**raph-based **A**gent **P**lanning (GAP), a novel framework that explicitly models inter-task dependencies through graph-based planning to enable adaptive parallel and serial tool execution. Our approach trains agent foundation models to decompose complex tasks into dependency-aware sub-task graphs, autonomously determining which tools can be executed in parallel and which must follow sequential dependencies. This dependency-aware orchestration achieves substantial improvements in both execution efficiency and task accuracy. To train GAP, we construct a high-quality dataset of graph-based planning traces derived from the Multi-Hop Question Answering (MHQA) benchmark. We employ a two-stage training strategy: supervised fine-tuning (SFT) on the curated dataset, followed by reinforcement learning (RL) with a correctness-based reward function on strategically sampled queries where tool-based reasoning provides maximum value. Experimental results on MHQA datasets demonstrate that GAP significantly outperforms traditional ReAct baselines, particularly on multi-step retrieval tasks, while achieving dramatic improvements in tool invocation efficiency through intelligent parallelization. The project page is available at: https://github.com/WJQ7777/Graph-Agent-Planning.

## 1    Introduction

Recent advances in large language model (LLM)-based autonomous agents have demonstrated remarkable capabilities in complex problem-solving tasks[1–6], ranging from scientific research and code generation to interactive web navigation and data analysis. A key enabler of these capabilities is tool-augmented reasoning, where agents leverage external tools such as search engines, calculators, code interpreters, and APIs to extend their problem-solving capacity beyond the inherent limitations of parametric knowledge.

Current approaches to tool-augmented reasoning can be broadly categorized into two paradigms: multi-agent systems (MAS) and tool-integrated reasoning (TIR) models. Multi-agent frameworks orchestrate multiple specialized agents with distinct roles and tool sets to collaboratively solve complex tasks. These systems have shown impressive performance on benchmarks requiring sophisticated workflows, such as software development and scientific research. However, they suffer from criti-

cal limitations: (1) high computational overhead due to redundant inter-agent communication and complex orchestration mechanisms; (2) inability to learn from data, as the underlying LLMs are not specifically trained for multi-agent coordination; and (3) reliance on prompt engineering rather than native model capabilities to achieve multi-turn, multi-tool workflows.

In contrast, Tool-Integrated Reasoning (TIR) models represent an emerging paradigm that explicitly trains LLMs to incorporate tool usage into their reasoning process. Recent work such as Search-R1[7] and WebThinker[5] has demonstrated that end-to-end training of models to invoke tools (e.g., `<search>` functions) at appropriate reasoning steps significantly outperforms prompt-engineered approaches. The TIR framework naturally aligns with the ReAct paradigm[4], enabling models to follow a "think-act-observe" pipeline in an end-to-end manner. However, existing TIR methods are fundamentally limited to sequential reasoning trajectories. They execute one action at a time and thus fail to exploit opportunities for parallel tool execution when sub-tasks are independent.

To address these limitations, we introduce Graph-based Agent Planning Paradigm (GAP), a novel training paradigm that enables LLM-based agents to perform dependency-aware planning through explicit graph-based reasoning. Our key insight is that by training models to construct and reason over task dependency graphs, they acquire the capability to autonomously determine optimal execution strategies, thereby executing independent tools in parallel when possible and sequential ones when necessary. This approach combines the efficiency and learnability of TIR models with the expressive power of multi-agent coordination, without the overhead of actual multi-agent orchestration. Our main contributions are:

- We introduce GAP, a novel training paradigm for agent foundation models that incorporates dependency-aware task planning, enabling dynamic parallel and serial tool execution. To our knowledge, this is the first work to explicitly train LLMs for graph-based reasoning over task dependencies in tool-augmented settings.

- We design and curate a high-quality dataset of 7,000 graph-based planning traces from the Multi-Hop Question Answering (MHQA) benchmark, using GPT-4o to synthesize dependency-aware reasoning trajectories. We apply a rigorous filter mechanism, ensuring that training data emphasize dependency modeling.

- We demonstrate through extensive experiments across seven question-answering benchmarks that GAP achieves a 0.9% average performance improvement on multi-hop reasoning tasks over state-of-the-art baselines. Moreover, our method significantly enhances efficiency by reducing interaction turns by up to 33.4%, while decreasing response length by 24.9% and maintaining robust generalization to out-of-domain datasets.

Our work establishes graph-based dependency modeling as a critical direction for developing more efficient autonomous agents, bridging the gap between sequential TIR models and complex multi-agent coordination. Through extensive experiments on MHQA, we demonstrate that GAP achieves significant improvements over traditional ReAct baselines in both accuracy and efficiency.

## 2 Background

Complex task reasoning often requires structured decomposition, specialized capabilities, and external tool integration. We review two prominent paradigms that used in single agent:

**ReAct-style Tool-Using**    The ReAct-style approach, exemplified by ReAct[4], leveraged few-shot exemplars to guide an LLM to interleave reasoning traces and actions within a "Thought-Action-Observation" cycle. This framework augments LLMs with structured reasoning by interleaving *thought* steps $\tau_t \in \mathcal{T}$ for planning, *action* steps $a_t \in \mathcal{A}$ for tool use, and *observation* steps $o_t \in \mathcal{O}$ for outcome processing. The reasoning trajectory follows:

$$(\tau_1, a_1, o_1, \tau_2, a_2, o_2, ..., \tau_T) \tag{1}$$

where each thought $\tau_t$ conditions on the history $h_t = [\tau_{1:t-1}, a_{1:t-1}, o_{1:t-1}]$ to determine next action.

**Tool-Integrated Reasoning**    Tool-Integrated Reasoning (TIR) enhances LLMs' code reasoning capabilities by tightly coupling natural language reasoning with external tool execution environments[8–10]. This approach enables a single agent to leverage external tools $\mathcal{T} = \{t_1, t_2, ..., t_M\}$ by maintaining a global state $S_t$ and selecting tools via policy $\pi(t_k \mid S_t)$. After executing tool $t_k$, the agent

observes outcome $o_t \sim \mathcal{O}(t_k, S_t)$ and updates its state:

$$S_t = f(S_{t-1}, t_k, o_{t-1}) \tag{2}$$

where $S_t$ denotes the reasoning state, $t_k$ represents the selected tool, and $o_t$ captures tool execution outcomes.

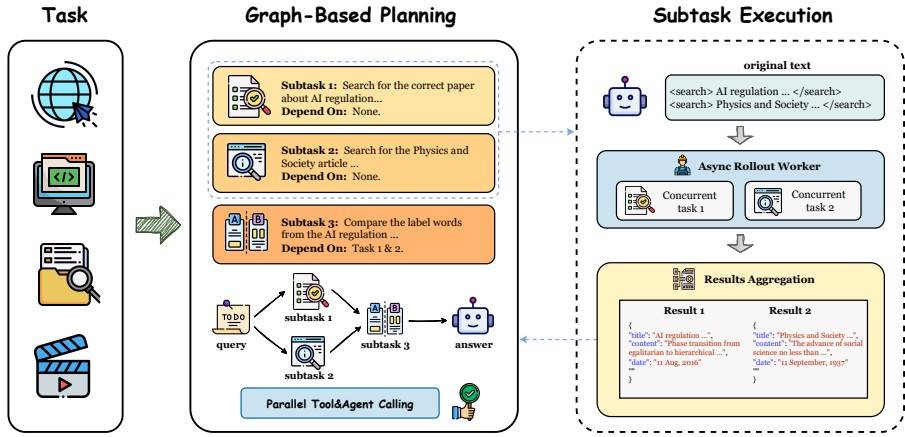

Figure 1: Illustration of Graph-based Agent Planning paradigm. GAP decomposes tasks into dependency-aware subtasks in the planning stage, enabling identification of parallelizable tool operations. The system supports parallel tool and agent calling for enhanced computational efficiency.

# 3 Graph-based Agent Planning Paradigm

In this section, we introduce the Graph-based Agent Planning (GAP) paradigm, a novel framework that enables LLM-based agents to perform dependency-aware reasoning and adaptive tool execution. We first formalize the problem setting (§3.1), then describe the core components of GAP including graph-based task decomposition (§3.2) and the dependency-aware execution strategies (§3.3). Figure 1 presents the complete GAP reasoning workflow, integrating task decomposition, graph construction, and adaptive execution.

## 3.1 Problem Formulation

We consider a task-solving scenario where an agent must answer a complex query $q$ by leveraging a set of external tools $\mathcal{T} = \{t_1, t_2, \ldots, t_n\}$. Each tool $t_i$ represents a specific capability, such as information retrieval (search), numerical computation (calculator), or code execution (python).

**Task Decomposition.** Given a complex query $q$, the agent must decompose it into a sequence of sub-tasks $S = \{s_1, s_2, \ldots, s_m\}$, where each sub-task $s_i$ requires invoking one or more tools from $\mathcal{T}$. The goal is to determine both which tools to invoke and when to invoke them.

**Dependency Graph.** We model task dependencies as a directed acyclic graph (DAG): $G = (V, E)$, where each vertex $v_i \in V$ represents a sub-task $s_i$ and each directed edge $(v_i, v_j) \in E$ indicates that sub-task $s_j$ depends on the output of sub-task $s_i$.

The absence of an edge between two vertices indicates independence, meaning those sub-tasks can be executed in parallel. The agent's objective is to construct this dependency graph and execute tools accordingly to maximize both efficiency and correctness.

## 3.2 Graph-based Task Decomposition

Unlike traditional sequential reasoning approaches (e.g., ReAct) that generate one action at a time, GAP explicitly constructs a task dependency graph during the planning phase. This process consists of three steps:

**Sub-task Identification.** The model first analyzes the input query $q$ and identifies the atomic sub-tasks required to solve it. For example, given the query "What are the populations of the capitals of France and Germany?", the model identifies four sub-tasks: $s_1$ retrieves the capital of France, $s_2$ retrieves the capital of Germany, $s_3$ retrieves the population of $s_1$'s result, and $s_4$ retrieves the population of $s_2$'s result.

**Dependency Analysis.** The model then reasons about dependencies between sub-tasks by analyzing their input-output relationships. A sub-task $s_j$ depends on $s_i$ if and only if $s_j$ requires the output of $s_i$ as input. In the example above, $s_3$ depends on $s_1$ as it needs to know Paris before querying its population, and similarly $s_4$ depends on $s_2$ as it needs to know Berlin. However, $s_1$ and $s_2$ are independent and can be executed in parallel, as are $s_3$ and $s_4$ given their respective dependencies are satisfied.

**Graph Construction.** Based on the dependency analysis, the model constructs the dependency graph $G$. We represent this graph using an adjacency structure that explicitly encodes:

```
Graph G:
  Nodes: [s1, s2, s3, s4]
  Edges: [(s1, s3), (s2, s4)]
  Parallel Groups: [{s1, s2}, {s3, s4}]
```

The model outputs this graph structure in a structured format that enables downstream execution planning. We train the model to generate this representation using a special token sequence:

```
<graph>
<node id="s1">search("capital of France")</node>
<node id="s2">search("capital of Germany")</node>
<node id="s3" depends="s1">search("population of {s1}")</node>
<node id="s4" depends="s2">search("population of {s2}")</node>
</graph>
```

### 3.3 Dependency-Aware Execution Strategies

Given the constructed dependency graph $G$, GAP determines an optimal execution strategy that balances parallelization opportunities with dependency constraints. We formalize this as a scheduling problem.

**Execution Levels.** We partition the graph $G$ into execution levels $L_0, L_1, \ldots, L_k$ using topological sorting, where:

- Level $L_0$ contains all nodes with no incoming edges (independent initial tasks)
- Level $L_i$ (for $i > 0$) contains nodes whose dependencies are all in levels $L_0, \ldots, L_{i-1}$

All sub-tasks within the same level $L_i$ can be executed in parallel, as they have no dependencies on each other.

**Parallel Execution.** For sub-tasks in the same execution level, the model generates a parallel tool call batch:
$$\text{Batch}_i = \{(t_j, \text{args}_j) \mid s_j \in L_i\}$$
where $t_j$ is the tool selected for sub-task $s_j$ and $\text{args}_j$ are its arguments. All tools in $\text{Batch}_i$ are invoked simultaneously, and the model waits for all results before proceeding to the next level. In Algorithm 1, we demonstrate the reasoning process of our proposed method.

## 4 Training Pipeline

### 4.1 Data Synthesis

During the Supervised Fine-Tuning (SFT) stage, we generate Graph-based Action Planning (GAP) trajectories using our proprietary multi-agent system. This approach is inspired by the multi-agent

**Algorithm 1** Graph-based Agent Planning with Parallel Tool Execution

---

**Require:** Input query $x$, policy model $\pi_\theta$, tool set $\mathcal{T}$, maximum turns $B$
**Ensure:** Final answer $y$
 1: Initialize rollout $y \leftarrow \emptyset$, turn count $b \leftarrow 0$
 2: **// Phase 1: Planning**
 3: Generate $y_{\text{plan}} \sim \pi_\theta(\cdot \mid x, y)$ until `</plan>`
 4: Parse dependency graph $G = (V, E) \leftarrow \text{ParseGraph}(y_{\text{plan}})$
 5: Compute execution levels $\{L_0, \ldots, L_k\} \leftarrow \text{TopologicalSort}(G)$
 6: $y \leftarrow y + y_{\text{plan}}$
 7: **// Phase 2: Level-wise Execution**
 8: **for** each level $L_i$ and $b < B$ **do**
 9:     Generate $y_b \sim \pi_\theta(\cdot \mid x, y)$ until `</tool>`
10:     $y \leftarrow y + y_b$
11:     **if** `<tool>` detected in $y_b$ **then**
12:         Extract queries $\{q_j\}_{j=1}^{|L_i|} \leftarrow \text{Parse}(y_b)$
13:         Execute in parallel: $\{o_j = \mathcal{T}(q_j)\}_{j=1}^{|L_i|}$
14:         $y \leftarrow y + $ `<observation>`$[o_1, \ldots, o_{|L_i|}]$`</observation>`
15:         $b \leftarrow b + 1$
16:     **end if**
17: **end for**
18: **// Phase 3: Synthesis**
19: Generate $y_{\text{ans}} \sim \pi_\theta(\cdot \mid x, y)$ until `</answer>`
20: **return** $y + y_{\text{ans}}$ =0

---

distillation framework proposed by Chain-of-Agents[11]. Starting with the Natural Questions (NQ) [12] and HotpotQA [13] datasets, we employ GPT-4o as the backend model to simulate the graph-based planning process. The prompt template refers to Section B.

To ensure the quality of the GAP training, we implemented a filtering process to select only high-quality, non-trivial trajectories from the varied data sources. We apply three key filtering criteria to curate the training data:

*(1) Complexity threshold:* We remove samples that can be completed with fewer than 3 search operations, as such trajectories are overly simplistic and do not benefit from parallel retrieval strategies.

*(2) Task diversity:* We maintain a 6:4 ratio between samples utilizing parallel retrieval and those using sequential retrieval, ensuring the model's generalization capability across different retrieval patterns.

*(3) Length constraint:* We filter out excessively long samples, retaining only those within approximately 2000 tokens. Overlong samples typically indicate missing relevant content in the offline dataset rather than genuine retrieval difficulty, and such redundant samples are detrimental to training efficiency, particularly given our objective of minimizing redundancy and maximizing retrieval efficiency.

Following this pipeline, approximately 7,000 high-quality training trajectories were generated through trajectory synthesis and quality filtering.

## 4.2 Supervised Fine-tuning for Cold Start

We fine-tuned the Qwen2.5-3B-Instruct model on our filtered dataset. The model learns to internalize graph-based planning strategies, enabling it to solve tasks by leveraging graph representations. The training objective minimizes:

$$\mathcal{L}_{\text{SFT}} = -\sum_{i \notin \mathcal{O}} \log \pi_\theta(\tau_i | \tau_{<i}, \mathbf{q})$$

with observation masking ($\mathcal{O}$) to prevent environmental noise propagation. This establishes robust cold start for downstream RL.

### 4.3 End-to-End Agentic Reinforcement Learning

While supervised training establishes a baseline understanding of parallel execution, it merely guides the model to imitate the provided demonstrations, and does not optimize computational efficiency or reasoning effectiveness. We further fine-tune the language model with fully end-to-end reinforcement learning. During RL-based finetuning, we iteratively sample reasoning traces from our current policy, assign them a reward according to the correctness of the proposed solution, and optimize policy parameters with DAPO[14]. In this stage, the model learns to strategically determine when, how, and how broadly to invoke child threads, maximizing performance by balancing the trade-offs between parallel exploration and the context window constraint. We use the VeRL framework[15] for DAPO training.

**Reward function**   Reward signals are critical for shaping RL dynamics in open-ended web agent tasks. Our framework adopts a graph-based design, built on two key considerations: Format consistency is inherently ensured through high-quality supervised fine-tuning and effective cold-start, obviating the need for explicit format validation rewards. For evaluating answer correctness, we use rule-based metrics to provide binary assessments. Our reward function is:

$$\mathcal{R}_{\mathrm{acc}}(\tau) = score_{\mathrm{answer}} \tag{10}$$

where $score_{\mathrm{answer}} \in \{0, 1\}$ is 1 if the final prediction is correct. Future work could productively explore multi-objective reward formulations that incorporate auxiliary signals.

## 5   Experiments

### 5.1   Setup

**Datasets**   We select seven benchmark datasets that encompass a diverse range of search with reasoning challenges by following the setup of [7]. These datasets are categorized as follows: (1) General Question Answering: NQ[12], TriviaQA[16], and PopQA[17]. (2) Multi-Hop Question Answering: HotpotQA[13], 2WikiMultiHopQA[18], Musique[19], and Bamboogle[20]. Following [7], we merge the training sets of NQ and HotpotQA as the training data and conduct evaluations on the validation or test sets.

**Metrics**   We use Exact Match (EM) as the evaluation metric to assess both in-domain and out-of-domain performance. In Figure 2, we follow [21] and adopt the cost-of-pass metric to quantify model efficiency. The cost-of-pass metric, denoted as $v(m, p)$, represents the expected monetary cost of using a model $m$ to generate a correct solution for a problem $p$. It is computed as the ratio of the cost of a single inference attempt, $C_m(p)$, to the success rate, $R_m(p)$:

$$v(m, p) = \frac{C_m(p)}{R_m(p)}$$

Here, the cost of a single inference attempt, $C_m(p)$, is defined as:

$$C_m(p) = n_{\mathrm{in}}(m, p) + n_{\mathrm{out}}(m, p)$$

where $n_{\mathrm{in}}(m, p)$ and $n_{\mathrm{out}}(m, p)$ are the number of input and output tokens for model $m$ on problem $p$, respectively. The success rate $R_m(p)$ is estimated by the proportion of correct responses. This metric represents the expected cost of using a model to generate a correct solution for a problem.

**Baseline**   We conduct comprehensive comparisons against state-of-the-art methods to evaluate our approach across MHQA datasets. We systematically evaluate a suite of tool-augmented methods, including Search-R1[7], ZeroSearch[22], StepSearch[23] and Chain of Agents[11].

**Implementation Details**   We conduct experiments using Qwen2.5-3B models (Yang et al., 2024) as the backbone of the agent, E5[24] as the embedding model, and 2018 Wikipedia dump[25] as the corpus. All experiments are conducted on 8 NVIDIA A100 GPUs.

Table 1: Performance comparison on various QA datasets, with Qwen2.5-3B-Instruct serving as the foundation model. **Bold** indicates best results among all methods. †/* denote in-domain/out-ofdomain datasets respectively.

| Methods | Single-Hop QA | | | Multi-Hop QA | | | |
|---|---|---|---|---|---|---|---|
| | NQ† | TriviaQA* | PopQA* | HotpotQA† | 2wiki* | Musique* | Bamboogle* |
| Qwen2.5-3B-Instruct | 10.5 | 13.2 | 18.8 | 9.9 | 20.2 | 4.7 | 1.2 |
| Search-R1 | 38.3 | 59.3 | **43.5** | 37.6 | 31.7 | 15.1 | 37.1 |
| ZeroSearch | **43.3** | **61.6** | 41.4 | 27.4 | 30.0 | 9.8 | 11.1 |
| StepSearch | - | - | - | 34.5 | 32.0 | 17.4 | 34.4 |
| AFM-RL-3B | 39.3 | 58.2 | 42.4 | 41.1 | 39.8 | **19.0** | 43.2 |
| *GAP-3B (Ours)* | 39.6 | 59.1 | 40.1 | **42.5** | **41.7** | 18.7 | **43.8** |

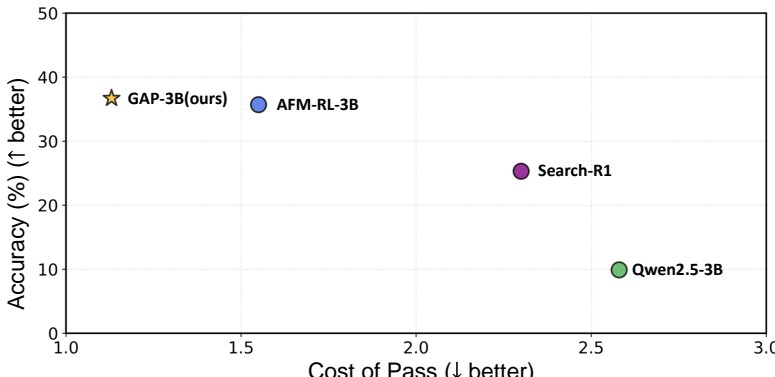

Figure 2: Performance-cost trade-off comparison across different models on HotpotQA. GAP-3B achieves the best balance with highest accuracy at lowest cost.

## 5.2 Results and Efficiency Analysis

Table 1 presents comprehensive results comparing GAP against baseline methods across seven benchmarks using four model configurations. Beyond accuracy improvements, GAP demonstrates significant efficiency gains on multi-hop reasoning tasks through parallel decomposition of independent sub-queries. As shown in Table 2 and Figure 3, our method achieves superior performance across multiple efficiency metrics compared to sequential baselines. Figure 2 further illustrates this advantage through a performance-cost trade-off analysis on HotpotQA. Our analysis reveals several key findings:

**Superior performance on complex multi-hop reasoning.** Our method demonstrates particular strength on multi-hop benchmarks, outperforming the best baseline by 0.9% on average across four multi-hop datasets (HotpotQA, 2Wiki, Musique, Bamboogle). This indicates that GAP successfully learns strategies for decomposing and parallelizing complex queries. On single-hop questions, GAP achieves comparable performance to ZeroSearch, which trains an LLM to simulate search engines and generate pseudo-context. Compared to Search-R1, our method shows a substantial 3.95% improvement.

**Reduced interaction turns and faster execution.** Compared to Search-R1, which retrieves information via sequential query generation, GAP significantly reduces the number of LLM interaction turns. On HotpotQA, GAP requires only 1.78 turns compared to Search-R1's 2.27 turns (21.6% reduction), while on 2Wiki, the reduction is even more pronounced (2.03 vs. 3.05 turns, 33.4% reduction). The cumulative distribution functions in Figure 3 further illustrate this advantage: our method efficiently responds to questions within 2 turns in most cases, whereas Search-R1 typically requires 3-6 turns. This reduction in interaction turns directly translates to faster execution times, with GAP achieving 32.3% and 21.4% time cost reductions on HotpotQA (168 vs. 248s) and 2Wiki (206s vs. 262s), respectively. Notably, the model autonomously determines parallelizability based on learned patterns during inference, demonstrating strong generalization ability.

**Shorter response length and lower deployment cost.** GAP also significantly reduces response length compared to baselines. As shown in Figure 3, Search-R1 generates substantially more tokens to support reasoning over retrieved documents, while GAP learns efficient reasoning strategies that reduce response length by 24.9% on HotpotQA (416 vs. 554 tokens) and 20.3% on 2Wiki (452 vs. 567 tokens). This reduction in generated tokens directly decreases deployment costs and increases throughput, which are critical factors for real-world applications. Furthermore, these efficiency gains generalize across domains: while HotpotQA is an in-domain dataset, similar improvements are observed on out-of-domain benchmarks, demonstrating that the learned parallel decomposition patterns transfer effectively to new scenarios. These results validate that GAP not only improves accuracy but also makes multi-hop reasoning more practical and cost-effective for deployment.

Table 2: Efficiency comparison on HotpotQA and 2wiki, with Qwen2.5-3B-Instruct serving as the backbone. **Time cost** refers to the time required to infer a batch of data. **Bold** indicates best results among all methods. †/* denote in-domain/out-ofdomain datasets respectively.

| HotpotQA† | Acc↑ | Length↓ | Time Cost(s)↓ | # Turns↓ |
|---|---|---|---|---|
| *Qwen2.5-3B-Instruct* | *9.9* | *256* | *114* | *1.11* |
| Search-R1 | 25.3 | 584 | 221 | 2.69 |
| AFM-RL-3B | 35.7 | 554 | 248 | 2.27 |
| *GAP-3B (Ours)* | **36.7** | **416** | **168** | **1.78** |

| 2wiki* | Acc↑ | Length↓ | Time Cost(s)↓ | # Turns↓ |
|---|---|---|---|---|
| *Qwen2.5-3B-Instruct* | *10.5* | *277* | *121* | *1.12* |
| Search-R1 | 31.7 | 651 | 254 | 3.05 |
| AFM-RL-3B | 39.8 | 567 | 262 | 2.64 |
| *GAP-3B (Ours)* | **41.7** | **452** | **206** | **2.03** |

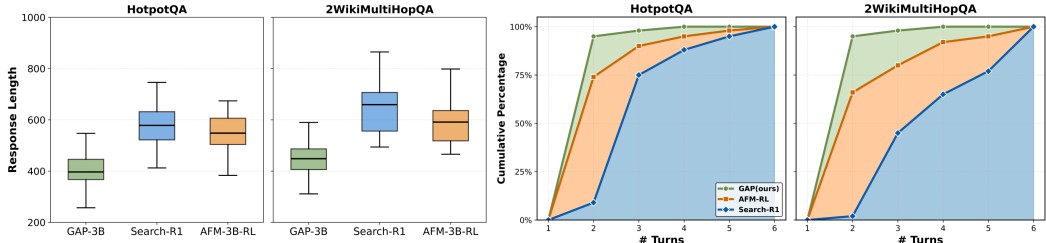

Figure 3: Illustration of total turns and response length on HotpotQA and 2WikiMultiHopQA datasets. Left panels show response length distribution, right panels show cumulative percentage of problems solved within different numbers of turns.

## 6 Conclusion

In this paper, we introduced GAP (Graph-based Agent Planning), a novel paradigm that enables LLM-based agents to perform dependency-aware reasoning and adaptive tool execution. By explicitly modeling task dependencies through graph-based planning, GAP addresses the fundamental limitation of sequential execution in existing approaches like ReAct, achieving significant improvements in both efficiency and accuracy. Our key contribution lies in training agent foundation models to decompose complex queries into dependency graphs, autonomously determining which tools can be executed in parallel and which must follow sequential dependencies. Through a carefully designed two-stage training strategy, we demonstrate that GAP substantially outperforms traditional sequential baselines, particularly on multi-step retrieval tasks requiring sophisticated reasoning.

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

## A  Related Work

### A.1  Tool-Integrated Reasoning Method

Training Large Language Models for multi-turn Tool-Integrated Reasoning (TIR) represents a promising frontier in Reinforcement Learning. Representative works such as ARPO[26], SimpleTIR[9], and ToRL[8] adopt similar strategies: models are post-trained with SFT or RL, and outputs are structured (e.g., ...) to trigger tool execution, feeding results back into the reasoning loop. Some extend RL-based Tool-Integrated Reasoning by improving small LLMs' tool-use capability, stabilizing multi-turn reasoning, and rewarding tool-use sequences independent of final answers.

Today, such tool-integrated reasoning is no longer a niche capability but a baseline feature of advanced agentic models. Mature commercial and open-source systems, such as OpenAI's DeepResearch and o3[27], Kimi K2[28], Microsoft rStar2-Agent[29] and Meituan LongCat[30], routinely incorporate these RL-honed strategies, underscoring the centrality of outcomedriven optimization in tool-augmented intelligence. Recent work theoretically proves that TIR fundamentally expands LLM capabilities beyond the "invisible leash" of pure-text RL by introducing deterministic tool-driven state transitions, establishes token-efficiency arguments for feasibility under finite budget.

### A.2  Agent Foundation Model

The development of Agent Foundation Models (AFMs) marks a pivotal shift towards building models with innate reasoning and tool-use capabilities. A significant insight driving this field is that exceptional agentic performance is not solely dependent on model scale. Recent pioneering works, notably Chain-of-Agents[11] and Cognitive Kernel-Pro[31], have demonstrated that even models at smaller scales can achieve state-of-the-art agentic abilities when trained with rigorous, purpose-built paradigms.

These approaches address the limitations of scale-dependent capabilities through two key innovations: sophisticated data synthesis and specialized agent-centric training. The Chain-of-Agents framework employs a process of multi-agent knowledge distillation and outcome-driven reinforcement learning. This teaches a single, smaller model to internally simulate the collaborative roles of a multi-agent team, enabling it to rival the performance of much larger models or complex systems on benchmarks like GAIA[32] and WebArena[33], but with dramatically improved inference efficiency.

Similarly, Cognitive Kernel-Pro demonstrates that a meticulously designed open-source framework, combined with a systematic methodology for generating high-quality, verifiable training data across various domains (web, file, code), can produce smaller models that compete with systems relying on massive proprietary APIs. Collectively, these works prove that the strategic focus on training quality and architectural innovation is a viable path to creating highly capable and practical agents, making advanced agentic intelligence more accessible and efficient.

## B  Case Study

We conduct case studies to gain a deeper understanding of the behavior and capabilities of our GAP framework. The results are shown in the following tables.

Table 3: Template of *GAP* framework on MHQA benchmark. question will be replaced with the specific question during training and inference.

You can respond to questions using the following 6 functions: think, plan, search, observation, reflection and answer. Function Descriptions:
1. think: Provide reasoning, justification, and synthesis of information before using other functions. Begin with <think> and end with </think>.
2. plan: Break down the question into sub-tasks with explicit dependencies. Format each task as - Task ID: unique identifier (Task1, Task2, etc.), Description: what to search/investigate, Dependencies: which tasks must complete first (use "none" if independent). Begin with `<plan>` and end with `</plan>`.
3. search: Execute search queries. For parallel searches, separate multiple queries with |. For single search, use one query. Begin with `<search>` and end with `</search>`.
4. observation: Results from search functions. Begin with `<observation>` and end with `</observation>`.
5. reflection: Evaluate progress and suggest plan modifications if needed. Begin with <reflection> and end with </reflection>.
6. answer: Final confident answer. Begin with `<answer>` and end with `</answer>`.
Execution Rules:
1. Always use think before other functions.
2. Use <plan> to create a task dependency graph.
3. Execute independent tasks in parallel by using | separator in <search>.
4. Use think to synthesize and analyze results from multiple searches.
5. Tasks with dependencies execute only after prerequisites complete.
6. Use reflection if the plan needs major adjustment.
7. Multiple final answers should be separated by |.
Question: question.

Table 4: Case trajectory of *GAP* framework on MHQA benchmark. We select a typical case of parallelizable multi-hop execution for reference.

Question: What occupation was shared by both John Frankenheimer and Tiffanie DeBartolo?
<think>This question requires finding information about two different people and identifying their shared occupation. I need to search for each person's career information and then compare them to find commonalities.</think>
`<plan>`
Task 1: Search for John Frankenheimer's occupations and career
- Dependencies: none
Task 2: Search for Tiffanie DeBartolo's occupations and career
- Dependencies: none
Task 3: Compare their occupations to identify shared ones
- Dependencies: Task 1, Task 2 `</plan>`
<think>Tasks T1 and T2 are independent and can be executed in parallel using the | separator in search. After getting both results, I'll analyze them to find the shared occupation.</think>
`<search>`John Frankenheimer occupation career director | Tiffanie DeBartolo occupation career director novelist`</search>`
`<observation>` Doc 1 - John Frankenheimer: John Frankenheimer (1930-2002) was an American film and television director. He was known for directing films such as "The Manchurian Candidate" (1962), "Seven Days in May" (1964), and "The French Connection II" (1975). He was primarily recognized as a director in both film and television industries.
Doc 2 - Tiffanie DeBartolo: Tiffanie DeBartolo is an American novelist and film director. She is the author of novels including "God-Shaped Hole" and "How to Kill a Rock Star." She also directed the independent film "Dream for an Insomniac" (1996). She works as both a writer and director. `</observation>`
<think>Now I can synthesize the results from both parallel searches: - John Frankenheimer: Film and television director - Tiffanie DeBartolo: Novelist and film director
The shared occupation between them is "director" - both have worked as film directors. John Frankenheimer was primarily known as a director, while Tiffanie DeBartolo works as both a novelist and director.</think>
`<answer>`director`</answer>`

