# OpenReview forum: "[Regular] GAP: Graph-based Agent Planning with Parallel Tool Use and Reinforcement Learning"
_NeurIPS.cc/2025/Workshop_Mexico_City/NORA — NeurIPS 2025 Workshop NORA Oral_

### Official Review · Reviewer_v3a6 · 2025-11-01
**Novel training paradigm and strong experimental section**

**Rating:** 8
**Confidence:** 3

**Review:**

Summary:
The paper proposes a novel training paradigm of explicitly training LLMs for graph-based reasoning in tool-augmented settings. GAP solves the problem of sequential execution of parallel subtasks in ReAct-style paradigms by modeling inter-task dependency through graph-based planning using a DAG. A finetuning dataset is generated by their proprietary multi-agent system. The supervised finetuning helps with cold start and reduces reliance on prompt engineering for tool-selection based workflows.


Pros:
1. The novelty is clear with a new training paradigm that parallelizes graph-based tool selection.
2. The experimental evidence is strong with several tool-selection based baselines being used, datasets, and efficiency analysis being included.
3. GAP not only has better accuracy in performance, but leads to reduced token consumption and shorter deployment lengths. This makes it highly suitable for real-world applications.

Cons:
1. This is a minor point, but more evaluation metrics for accuracy would be great to include.
2. More details on the graph construction algorithm will be helpful. What did you use to identify the parallel groups (seems like topological sort was used for identifying execution levels)? What approaches have you tried and what were the pros/cons in relation to the specific problem you are solving?
3. GAP does not have the overwhelmingly best performing in Table 1.

Questions:
1. What is the qualitative impact of in-domain or out-of-domain on the inference using GAP?

---

### Official Review · Reviewer_Paf9 · 2025-11-04

**Rating:** 6
**Confidence:** 3

**Review:**

This paper presents Graph-based Agent Planning (GAP), a novel framework for large language model (LLM)-driven autonomous agents that addresses the inefficiency of existing sequential tool-use paradigms like ReAct. The authors propose to represent inter-task dependencies through explicit graph structures, allowing the agent to identify which sub-tasks can be executed in parallel and which require serial execution. The framework combines supervised fine-tuning on a curated graph-planning dataset and reinforcement learning guided by correctness-based rewards. Experiments on Multi-Hop Question Answering (MHQA) benchmarks show significant improvements in both reasoning accuracy and execution efficiency, particularly in multi-step retrieval tasks.

Strengths:

The paper addresses an important limitation in LLM-based autonomous agents: the lack of parallel reasoning and tool execution. The proposed GAP framework is conceptually elegant and technically sound, offering a clear mechanism for dependency-aware task decomposition. The introduction of graph-based planning aligns well with cognitive task modeling and provides interpretability in reasoning structure. The two-stage training pipeline (SFT + RL) is thoughtfully designed and justified. Empirical results on MHQA benchmarks demonstrate strong performance gains, both in terms of accuracy and tool efficiency, supporting the effectiveness of the proposed approach.

Weaknesses:

While the conceptual motivation is strong, the paper could benefit from deeper theoretical analysis or ablation studies that isolate the contribution of each component, particularly the effect of graph structure learning versus parallel execution scheduling. The scalability of GAP to open-domain or real-world multi-tool environments is not fully explored, leaving open questions about generalization beyond MHQA tasks. Additionally, the paper would be strengthened by a qualitative analysis of failure cases or visualizations of learned dependency graphs to better illustrate interpretability claims. The reinforcement learning stage, while promising, lacks detail on stability and convergence, which may affect reproducibility.

---

### Official Review · Reviewer_WurN · 2025-11-05
**Interesting, but the result analysis/report could be improved**

**Rating:** 6
**Confidence:** 3

**Review:**

The paper proposes a framework called Graph-based Agent Planning (GAP) for enabling LLM-based agents to decompose complex tasks in to subtasks and organise them into a dependency graph. Each subtask is assumed to be solved by a single tool calling, and the dependency graph is used to determine what tools can be called in parallel. The paper describes how an LLM can be trained to follow the GAP approach in two steps: SFT, then reinforcement learning. Experiments on several open-source datasets show interesting results.

Main pros:
- While the general idea of the framework is quite simple, the specific algorithm and training choices are not straightforward. Moreover, the experiments compare several models on several datasets, which adds value to the paper.
- The paper is clear most of the time and easy to read.

Main cons:
- Not enough details are provided to reproduce the experiments. I wonder whether the authors intend to share their code?
- The reported results are strangely incomplete. Why is the cost of pass only reported for some datasets?
- The GAP framework seems to underperform on single-hop questions, and the gains on multi-hop questions are small.

Other questions and remarks:
- Line 105: I think that the authors mean "the absence of a path"
- I'm not sure to understand how is used the representation presented at line 125; moreover, the representation handled by the LLM (line 131) does not seem to contain the "Parallel Groups" information.
- Why partition the graph G into execution levels instead of just running tasks when all their dependencies are finished?
- How was the multi-agent system used to generate the plans for the SFT data (line 154)? I find the description a bit vague.
- Figure 2: I find a bit misleading (in favor of GAP-3B) to plot accuracy against cost of pass, since cost of pass is Cm/accuracy. I think accuracy against Cm would give a clearer picture.

---

### Official Review · Reviewer_dH86 · 2025-11-07
**An interesting novel approach to LLM agent planning enabling parallel tool use, but with some missing details and presentation issues.**

**Rating:** 7
**Confidence:** 3

**Review:**

The paper describes a new approach to agent based problem solving using LLMs and external tools. The approach uses a fine-tuned LLM to decompose problems into sets of sub-tasks that can be addressed with external tools, and to represent these sub-tasks as a DAG that encodes dependencies between the tasks. A topological sort of the graph is then used to determine which of the tasks can be solved in parallel. The system is fine-tuned on data derived from the Multi-Hop Question Answering benchmark, combined with a reinforcement learning step. The results demonstrate state-of-the-art performance on MHQA, but at a reduced cost per correct answer compared to sequential reasoning approaches.

Pros: The approach seems novel and successful, establishing a new state of the art on MHQA. It combines LLM-based multi-step reasoning with symbolic scheduling (topological sort on the task graph). People working on agentic LLMs will find this work valuable and interesting. I thought the using supervised fine-tuning with an objective that encodes correct "parallelization" of the tools, followed by reinforcement learning was clever.

Cons:
The paper lacks some details that limit replicability, especially in section 4. Generally, I found section 4 to be a bit technically dense. For example, the SFT objective could be explained in more detail (I think I understand how this works, but some symbols are not clearly defined).

It is unclear how exactly the dataset was generated. The paper just references a "proprietary multi-agent system". What does "The prompt template refers to section B" mean? I thought maybe this is a reference to the appendix, but appendix B contains a case study.

It would have been interesting to see how much RL contributes to the final performance, as opposed to just using fine-tuning by itself. Related, did you evaluate performance on the graph generation task independently? It looks like the paper only evaluates the end-to-end performance on the various QA tasks. Related, the paper claims that supervised fine-tuning ensures that the output graphs are formatted correctly (and I assume that means they must be DAGs), but is this really guaranteed in all cases?

Performance on single-hop tasks is lower than for some of the baseline systems. This makes some sense since the system was not specifically trained on these tasks, but I think this requires some discussion.

The paper does not contain a dedicated related work section -- it would have been interesting to place this work in the context not just of LLM agents, but also traditional (symbolic) approaches to scheduling and multi-step reasoning.

The paper mentions a few limitations in different places (such as the rather simple binary RL reward function), but it would have been nice to see a more targeted discussion of limitations and future work.

Some formatting comments:
Some of the figures are difficult to read (text too small), especially the legend in the two right panels in figure 3.
Generally, it is preferable to place tables and figures, such as Figure 1 (!), at the top of the page so as not to interrupt the flow of the text."

---

### Official Review · Reviewer_1cox · 2025-11-07
**GAP: Graph-based Agent Planning with Parallel Tool Use**

**Rating:** 4
**Confidence:** 4

**Review:**

This paper presents Graph-based Agent Planning (GAP). It is a framework to make LLM agents more efficient. The authors claim that current methods of agentic reasoning (like ReAct) are slow because they handle tasks sequentially. GAP solves this by training an LLM to first split the task into smaller pieces and create a dependency graph (via a topological ordering). This way independent tasks can be run in parallel.

To this the authors train a specialized LLM. They do SFT on a Gpt 4o created dataset of 7,000 plans, and then do RL to optimize for correctness. Experiments show that GAP significantly outperforms sequential methods on multi-hop question answering, improving both accuracy and efficiency by reducing interaction turns.

Reasons for acceptance:
- This paper addresses a very practical limitation of popular agent frameworks. Being able to parllelize tool invocation is very useful.
- Authors create and release a new dataset of 7,000 graph based planning traces.
- The empirical results are impressive, especially the efficiency gains. A 33% reduction in interaction turns and a 25% reduction in response length are significant. This shows the method isn't just a small tweak but provides a substantial real-world benefit in terms of speed and cost.

Reasons to reject:
- I'm not entirely sure how much of a good fit this paper is for this workshop. The workshop is explicitly about KGs and their interplay with agents. There is no KG used here but the paper is about task dependency graphs for planning.
- Comparison to ReAct feels a bit forced, since ReAct is designed to be sequential. A more powerful comparison would have been against a multi-agent framework where different agents can act in parallel, even if it's more complex to set up.
- The paper doesn't really discuss the main failure mode: what happens when the model generates an incorrect dependency graph? Creating a correct task graph is a hard planning problem in itself, and the paper seems to assume the LLM can just do this reliably.
- The scalability of the parallel execution is unclear to me. The examples in the paper (like finding capitals of two countries) are simple. It's not clear how the agent or the context window would handle a plan that requires, for example, ten parallel tool calls.

---

### Official Review · Reviewer_zWod · 2025-11-07
**An intuitive tool for multi-hop question anwering**

**Rating:** 7
**Confidence:** 4

**Review:**

The authors tackle the problem of the limitation of the sequential planning in the agentic system, and propose a dependency-aware graph-based planner. The explicit dependency modeling shows better execution efficiency and task accuracy in multi-hop question answering benchmarks.

Strengths:
- The graph-based dependency is intuitive, especially in cost and latency optimization. The authors also provided comprehensive analysis in evaluation in terms of reduction in the token length, latency, turns and cost. These improvements are particularly impactful in industry.
- In accuracy, it validates the hypothesis of improvement on multi-hop questions while still doing well on 1-hop questions.

Weaknesses:
- It could be better to provide more apple-to-apple comparison in ablation. It would be great to have a baseline with the sequential ReAct as direct comparison to the Graph-based ReAct in experiments. This might be implicit in other baseline methods, but would be helpful to present an apple to apple comparison in the same setup.
- The training data for SFT is generated using the proprietary multi-agent system. It would be great to ask the same system to generate sequential trajectory as comparison to understand whether the accuracy improvement comes from the knowledge distillation of a better agentic system or from the graph-based dependency design.